# Motor Pathophysiology Related to Dyspnea in COPD Evaluated by Cardiopulmonary Exercise Testing

**DOI:** 10.3390/diagnostics11020364

**Published:** 2021-02-21

**Authors:** Keisuke Miki

**Affiliations:** Department of Respiratory Medicine, National Hospital Organization Osaka Toneyama Medical Center, 5-1-1 Toneyama, Toyonaka, Osaka 560-8552, Japan; miki.keisuke.pu@mail.hosp.go.jp

**Keywords:** acidosis, breathing, cardiopulmonary exercise testing, dynamic hyperinflation, muscle, ventilation

## Abstract

In chronic obstructive pulmonary disease (COPD), exertional dyspnea, which increases with the disease’s progression, reduces exercise tolerance and limits physical activity, leading to a worsening prognosis. It is necessary to understand the diverse mechanisms of dyspnea and take appropriate measures to reduce exertional dyspnea, as COPD is a systemic disease with various comorbidities. A treatment focusing on the motor pathophysiology related to dyspnea may lead to improvements such as reducing dynamic lung hyperinflation, respiratory and metabolic acidosis, and eventually exertional dyspnea. However, without cardiopulmonary exercise testing (CPET), it may be difficult to understand the pathophysiological conditions during exercise. CPET facilitates understanding of the gas exchange and transport associated with respiration-circulation and even crosstalk with muscles, which is sometimes challenging, and provides information on COPD treatment strategies. For respiratory medicine department staff, CPET can play a significant role when treating patients with diseases that cause exertional dyspnea. This article outlines the advantages of using CPET to evaluate exertional dyspnea in patients with COPD.

## 1. Introduction

Globally, chronic obstructive pulmonary disease (COPD) was the third leading cause of death in 2018 [1], and countermeasures against COPD are required. The most frequent and important complaint of patients with COPD is exertional dyspnea, the pathophysiology of which is known to includes several etiological factors [2,3,4,5]. There can be no improvement in exercise tolerance and physical activity for patients without mitigating dyspnea, which is difficult to bear. Customized treatment suitable for the condition must be provided to alleviate exertional dyspnea after exploring the causes of COPD, given that COPD is a systemic disease with comorbidities [6,7] and that exertional dyspnea can have diverse etiologies.

This article discusses the motor pathophysiology of COPD, which is directly related to the treatment of exertional dyspnea, based on previous insights into how dyspnea in COPD can be evaluated using cardiopulmonary exercise testing (CPET) [8]. Briefly, two types of protocols are used in CPET: a maximum (symptom-limited) incremental exercise test, and a constant work rate exercise test. Numerous exertional parameters are included, all of which are calculated using the ventilation amount, and the concentration of oxygen and carbon dioxide during inspiration or expiration [8,9,10]. CPET facilitates diagnosis and determination of the patient’s exertional pathophysiological condition and can help in the choice of treatment, evaluation of the response to treatment, and determination of the prognosis [8,11]. Although the interpretation of data obtained from CPET sometimes requires a comprehensive understanding of this test method, CPET can quickly provide deep insights into pathophysiological responses to exercise, reflecting crosstalk between the functions of the cardiopulmonary system and peripheral muscles [10]. This article also examines how CPET can facilitate the implementation of treatment strategies to further alleviate exertional dyspnea, with a particular focus on respiratory patterns.

## 2. Exercise Tolerance and Exercise-Limiting Factors 

The forced expiratory volume in one second (FEV_1_) is an important indicator that correlates well with minute ventilation (*V’*_E_), but has a weak relationship with exercise tolerance [12]. This distinction reflects the fact that the ventilatory efficiency is related to wasted ventilation [13,14,15] and to the fact that the difference between inspired and expired oxygen concentrations (measured as the difference between average inspired oxygen concentration and average expired oxygen concentration) [9,10], not just the ventilation amount, is related to exercise tolerance. In this context, it is worth noting that oxygen uptake (*V’*_O2_) is determined using an equation that includes the product of *V’*_E_ and the difference between inspired and expired oxygen concentrations, and that the average expired oxygen concentration is dependent on the collective cardiac, pulmonary and muscular metabolism [9,10]. Although many factors define the daily activity level of patients with COPD, this activity level is often influenced by dyspnea. Predicting the level of daily activity based on resting pulmonary function alone is difficult because other exercise limiting factors, such as cardiovascular disorders and lower limb fatigue, are not evaluated. Evaluation of exercise tolerance by CPET indicates the level of daily activity more directly and is also useful in predicting the patient’s prognosis [16,17,18,19,20]. Adequate measures are needed, mainly for patients who can engage only in activities of approximately 3 metabolic equivalents (METs) (maximal oxygen uptake of 10.5 mL/min/Kg) [18,21,22]. The pathophysiology of exertional dyspnea can be understood in due course, including the exercise-limiting factors concerned with dyspnea treatment. 

Accurate evaluation during the CPET of leg fatigue, in addition to dyspnea during CPET, is important for determining the treatment plan. The evaluation must be performed while the patient is not talking, thereby ensuring that there is no interference with exhaled gas evaluation. The 10-point modified Borg Scale, with 0 corresponding to no dyspnea and 10 corresponding to maximal dyspnea or leg fatigue, is often used [23]. When dyspnea and leg fatigue are evaluated during CPET, dyspnea alone, leg fatigue alone, or the combination of both dyspnea and leg fatigue is commonly reported at the end of exercise. On this scale of dyspnea, scores of 2 or 3 correspond to the anaerobic metabolism threshold and are associated with an elevated threshold of sympathetic nerve activity not only in COPD but also in interstitial pneumonia [3,24]. Therefore, it is useful to instruct people to engage, as part of their self-range, in activities for which dyspnea is 2–3 points on the modified Borg scale, thereby avoiding excessive exercise in daily life. If the causes of dyspnea are examined closely, the exercise-limiting factors among these causes can be classified broadly into three categories, namely: dyspnea due to ventilatory impairment, lower limb fatigue, and cardiovascular impairment (Figure 1).

### 2.1. Exertional Dyspnea Due to Cardiovascular Disorders

In previous studies of patients with COPD who exhibit exertional dyspnea [12,25], 10–17% of patients with exertional dyspnea were subjects whose dyspnea was due to cardiovascular disorders; many of these patients were observed to have abnormalities as assessed by electrocardiogram (ECG). Reports from the United States and Europe indicate that the percentage of ischemic heart disease and heart failure is higher among patients with COPD complications [26]. According to reports from Japanese cardiovascular facilities, the percentage of patients with COPD who also exhibited cardiovascular diseases represented 27% of individuals with COPD complications [27]. Determining that the heart is the limiting factor for exercise may be difficult if the ECG does not indicate an abnormality. In such cases, limitation due to the heart is indicated if (during CPET) the slope of *V’*_O2_ versus work rate (measured in watts) is reduced (Figure 1). Alternatively, a plateau phenomenon of the oxygen pulse (*V’*_O2_/heart rate (HR)), which is almost equivalent to the stroke volume, may be occurring (Figure 1). In either instance, a steep slope of HR versus *V’*_O2_ can be used as a reference to assess whether cardiovascular disorders are limiting factors for exercise (Figure 1) [28]. If the oxygen pulse plateaus during exercise, but there are no abnormalities in the ECG during exercise, then the pathological condition may be clarified using an echocardiogram to test for the existence of a valve disease, possibly including that of the mitral valve [29]. In addition, evaluating impairments of the pulmonary microvasculature may be informative, given that such impairments already can be seen in mild-stage COPD; indeed, reduced pulmonary blood flow during exercise has been reported in such patients [30,31]. Furthermore, the evaluation of sympathetic activity during exercise may provide important information on the pathophysiologic conditions underlying not only cardiovascular disease but also COPD during exercise (Figure 1). Elevated sympathetic activity already will be apparent in the resting condition in patients with advanced COPD, and the change of sympathetic activity has been shown to correlate with exertional dyspnea in patients with stable COPD [3,20,22]. Therefore, respiratory medicine department staff need to remember that cardiovascular diseases may be a cause of exertional dyspnea.

### 2.2. Measures and Treatment of Lower Limb Fatigue

Work by the author and colleagues has revealed that, in 20% of the patients with exertional dyspnea as the chief complaint, lower limb fatigue, not dyspnea, was actually the exercise-limiting factor (Figure 1); in such patients, resting pulmonary function was relatively preserved [12]. With the progression of respiratory or cardiovascular diseases, ventilatory disorders, circulatory disorders, or muscle sympathetic overactivity develop in addition to the lower limb fatigue, leading to the progression of dyspnea as the pathological condition worsens. Furthermore, aside from the case where dyspnea is the only exercise-limiting factor, if dyspnea and lower limb fatigue are of approximately the same intensity, then ventilation must compensate for exercise-induced acidosis (due to lactic acid production by the muscles). In such cases, exercise therapy focusing on the lower limbs may improve exertional dyspnea by suppressing excessive lactic acid production and consequently lowering the need for ventilation [32,33]. In any case, exercise therapy for the lower limbs should be performed from an early stage. The staff of respiratory medicine or cardiovascular departments often tend to neglect exercise therapy for the lower limbs in favor of their department’s respective specialty. Increasing physical activity at an earlier stage of COPD is associated with a better prognosis, and it is important to teach exercise habits that make use of hobbies from an earlier stage, thereby promoting behavioral changes [34]. Unfortunately, however, as COPD advances, the related functional impairments lead to muscle weakness and body weight loss [35,36]. Indeed, loss of muscle mass and muscle strength are greater in patients in the advanced stages of COPD [37], the progress of which particularly affects the lower limbs [36,38]. In patients that are underweight or have sarcopenia, exertional dyspnea can become very severe, because such patients must compensate for muscle impairments as well as for ventilatory impairments [39,40,41]. To treat such patients, it may be important to choose suitable therapies based on cardiopulmonary-peripheral muscle crosstalk. Ghrelin, first discovered in 1999 as a novel growth-hormone-releasing peptide isolated from the stomach [42], has a variety of effects, such as causing a positive energy balance and weight gain by decreasing fat utilization [43], stimulating food intake [44], and inhibiting sympathetic nerve activity [45]. The author and colleagues reported that, in cachectic patients with COPD, ghrelin administration with exercise training provided improvements in exertional dyspnea [39,46], respiratory strength [39], and exertional intolerance [47] in randomized, double-blind, placebo-controlled trials. 

In the future, as the population ages, the number of patients who are immobilized due to lower limb fatigue is expected to increase, and the need for exercise therapy of the lower limbs based on the cardiopulmonary-peripheral muscle crosstalk will increase further. CPET will be useful for identifying patients who are candidates for exercise therapy of the lower limbs.

### 2.3. Exertional Dyspnea Due to Ventilatory Impairment

In a study by the author and colleagues [12], exertional dyspnea due to ventilatory impairment accounted for 70% of patients with exertional dyspnea as the chief complaint (Figure 1). Here, the author considers the mechanism of exertional dyspnea due to ventilatory impairment and the relevant remedies that focus on respiratory patterns. Although tachypnea [48] is often considered to be a cause of dyspnea and a target of treatment, tachypnea is observed during maximal exercise in patients with COPD who exhibit a preserved exercise tolerance, and even in healthy individuals [49], and is considered to be a standard physiological mechanism used by the body to increase the amount of ventilation. On the other hand, it should be noted that, among patients with COPD who exhibit exercise intolerance, a surprisingly large number of subjects have a slow-shallow pattern with prolonged expiration (Figure 2a–c), where expiratory tidal volume (*V*_T_ex) is reduced without an increase in the respiratory frequency. Typically, such patients do not demonstrate a rapid-shallow pattern, where a rise in the *V*_T_ex is limited and is compensated by tachypnea [12]. The worsening of these mechanical ventilatory abnormalities during exercise is explained by i) high elastic load, ii) decreased dynamic lung compliance, and iii) increased resistive load of respiratory muscles, all of which lead to dynamic lung hyperinflation in COPD [50,51,52,53,54]. Studies of dynamic blood gas show that, in healthy individuals and patients with COPD who retain ventilation capacity, metabolic acidosis (resulting from elevated levels of lactic acid) progresses during exercise; ventilation compensation for such acidosis is detected as a decrease in bicarbonate ions, and exercise is terminated when metabolic acidosis is no longer compensated [12]. On the other hand, in patients with poor ventilation compensation capacity and decreased exercise tolerance, exercise is terminated when the patient develops respiratory acidosis (Figure 2d), which does not lead to an elevation in lactic acid levels but rather to an elevation in the partial pressure of arterial carbon dioxide and bicarbonate ions [12]. These observations explain why patients with dynamic lung hyperinflation [55,56] may become breathless. Dyspnea is an exercise-limiting factor in a high percentage of these patients [12], and an increase in ventilation capacity, ideally by increasing *V*_T_ex, or using a treatment to improve ventilatory efficiency, is expected to yield successful results. Interestingly, in patients with COPD who possess various resting pulmonary functions, ventilation limitations, and exercise tolerances, exercise was terminated when exercise-induced acidosis (pH) and dyspnea during maximal exercise (Figure 2e) were comparable [12]. This observation suggests that, though dyspnea is caused by complex factors, including the central nervous network and peripheral muscles [5], one of the common mechanisms in exertional dyspnea involves a compensatory mechanism that maintains acid-base homeostasis in the blood [3,12,57]. In the body, CO_2_ transport is affected by the exertional pH change that accompanies CO_2_ production due to ventilatory impairments or lactate production during exercise. Furthermore, in a study examining exercise-limiting factors in patients with idiopathic pulmonary fibrosis and COPD [3,58], exercise-induced acidosis was a limiting factor for exercise regardless of the concentration of oxygen administered. In addition, exertional hypoxemia was not a normal feature in heathy subjects who also felt exertional dyspnea at the end of exercise [24,59]. In other words, exertional dyspnea is associated with exercise-induced acidosis rather than with hypoxia, and ventilation may be an important compensatory mechanism for maintaining acid-base homeostasis, the mechanism of which may lead to optimal exercise performance. Therefore, adequate ventilation, especially exhalation, is fundamental to the treatment of COPD. As mentioned above, the slow-shallow pattern during exertion in patients with exercise intolerance is often accompanied by prolonged expiration (Figure 2c). Laveneziana et al. [60] reported that expiratory muscle activity in COPD was relatively increased during exercise but did not mitigate dynamic lung hyperinflation. More recently, it has been reported that expiratory muscle strength often increases in patients with COPD, perhaps to compensate for inadequate ventilation [61]; a negative correlation has been observed between maximal expiratory muscle strength at rest and maximal oxygen uptake, especially in patients with COPD who have prolonged expiration [61,62]. In other words, although the slow-shallow pattern prolongs expiration (Figure 2a–c), this ventilation pattern requires high expiratory muscle strength but does not raise oxygen uptake, making breathing difficult for the patient. Inadequate exhalation leading to prolonged expiration results in a large amount of air remaining in the lung after expiration (*V*_T_in-*V*_T_ex), expressed as the difference between the inspiratory tidal volume (*V*_T_in) and *V*_T_ex [61,62]. Excess expiratory muscle recruitment might be a compensatory mechanism to improve exercise intolerance. Further studies are necessary to clarify the implications for the COPD of excess expiratory muscle recruitment. 

#### 2.3.1. Improving Ventilatory Impairments to Reduce Exertional Dyspnea

Reducing the air remaining in the lung after the expiration of each exhaled breath is expected to improve dynamic hyperinflation, respiratory acidosis, and eventually dyspnea. COPD is recognized as a disease that primarily involves pulmonary parenchyma and obstructs the peripheral respiratory tract, which affects the exertional dyspnea [63]. Interestingly, however, collapse of the respiratory tract during forced expiration was reported in the 1960s among patients with COPD [64]. In addition, since its first description in the 1980s, exercise-induced laryngeal obstruction has been considered problematic because this condition may affect young adults and can mimic exercise-induced asthma [65,66,67,68]. Although there is no standardized methodology for confirming exertional laryngeal obstruction and dyspnea severity, continuous laryngoscopy during CPET has been reported to improve diagnostic sensitivity [69]. Furthermore, Baz et al. [70] recently reported that the central respiratory tract outside the mediastinum, namely, the vocal cords, is obstructed during exercise. As the obstruction intensifies, the degree of prolonged expiration also increases. The expiratory airflow limitation in patients with COPD involves obstruction of the peripheral respiratory tract and of the central respiratory tract, including the vocal cords, which affects exertional dyspnea and breathing patterns. In a preliminary study, the author and colleagues observed that expiratory pressure load training in patients with severe and very severe COPD increased the expiratory tidal volume, reduced the air remaining in the lung after expiration, and improved prolonged expiration. In other words, the slow-shallow pattern with prolonged expiration improved, which in turn resulted in improvements in subjective symptoms and exercise tolerance [62]. In contrast, inspiratory pressure load training, which is also referred to as inspiratory muscle training, has been recommended as a pulmonary rehabilitation (PR) program [71]. However, at least in patients with advanced COPD, no large studies have reported the adjunctive effects of inspiratory pressure load training added to PR [72,73,74]. In addition, Yamamoto et al. reported that, especially in underweight patients with advanced COPD, inspiratory pressure load training might lead to tachypnea and wasted ventilation, which would in turn decrease exercise performance; however, this point was made as part of a case report [75]. Further studies are needed to evaluate the effect of expiratory or inspiratory pressure load training on exertional dyspnea in patients with COPD. In the future, the author hopes to develop therapies for the slow-shallow pattern with prolonged expiration, which is the cause of exertional dyspnea in patients with advanced COPD.

#### 2.3.2. Reducing Ventilatory Demand to Reduce Exertional Dyspnea

In contrast to providing adequate ventilation, reducing ventilatory demand or increasing oxygen utilization may be of help in improving exertional dyspnea in COPD. Especially in patients who exhibit exercise intolerance due to the reduced capability to increase ventilation during exercise, a strategy related to oxygen utilization may be useful in reducing exertional dyspnea. Acupuncture, an Eastern medical practice, has been reported to improve exercise intolerance, dyspnea, and quality of life in patients with COPD [76,77]. In addition, the physiological benefits of acupuncture have been reported to include the relaxation of muscle tension, along with improvements of muscle/anti-muscle fatigue, muscle blood flow, and sympathetic control [78,79]. However, to date, little is known about the mechanism whereby acupuncture improves exertional dyspnea. Using CPET, Maekura et al. [80] investigated the effect and mechanism of acupuncture on exercise intolerance and exertional dyspnea in patients with COPD. Their findings demonstrated that the effects of acupuncture on exertional dyspnea were associated primarily with improved oxygen utilization and reduced ventilation during exercise. 

A similar mechanism explains how PR improves exercise performance and exertional dyspnea [81,82,83,84]. Using CPET, the author and colleagues [83] investigated how PR reduces exertional dyspnea; the results demonstrated that the reduced exertional dyspnea obtained from PR was associated with reduced ventilatory demand due to the economized oxygen requirements. In addition, although the mechanisms underlying meditative movement (tai chi, yoga, and qigong) on COPD are unclear, a systematic review and meta-analysis reported that the application of meditative movement as non-conventional therapies might improve exercise capacity, dyspnea, and health-related quality of life in COPD patients [85].

## 3. Conclusions

Diverse exertional dyspnea is related to the crosstalk between the heart, lungs, and muscles; further exploration of the exertional dyspnea patterns, which are related to cardiovascular disorders, ventilatory impairment, and/or lower limb fatigue, will not only facilitate elucidation of the dynamic pathophysiology of COPD but will also contribute directly to the treatment of patients with this disease. Reducing the air remaining in the lungs after the expiration of each exhaled breath is expected to improve dynamic hyperinflation, respiratory acidosis, and eventually dyspnea. Furthermore, reducing ventilatory demand or increasing oxygen utilization may also facilitate improvements in exertional dyspnea in COPD. Considering that CPET can provide key information on specific dysfunctions in COPD patients that can be used to help them maintain daily living activities and allow them to feel that they can “walk with a little more ease”, staff providing respiratory care should make the most of CPET as a more approachable test.

## Figures and Tables

**Figure 1 diagnostics-11-00364-f001:**
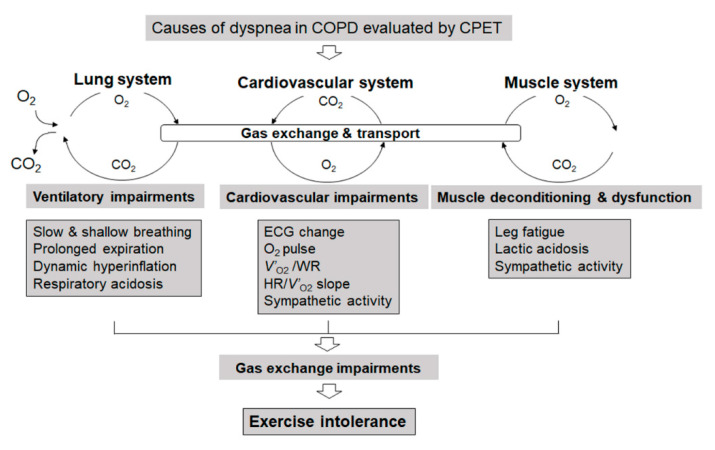
Schematic pathophysiologic pathway exertional dyspnea in chronic obstructive pulmonary disease. CO_2_: carbon dioxide; CPET: cardiopulmonary exercise testing; ECG: electrocardiogram; HR: heart rate; O_2_: oxygen; O_2_ pulse: *V’*_O2_/heart rate; *V’*_O2_: oxygen uptake; WR: work rate. This is an original figure (no permission is required).

**Figure 2 diagnostics-11-00364-f002:**
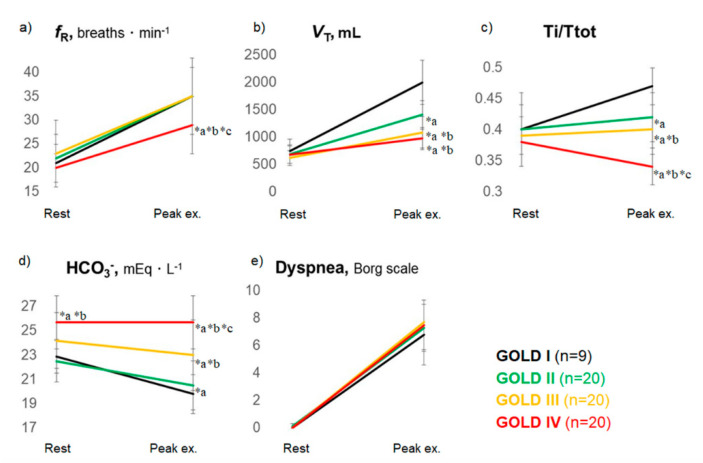
Typical responses to incremental exercise by patients with chronic obstructive pulmonary disease. Data are presented as mean ± SD. ex.: exercise; *f*_R_: respiratory frequency; HCO_3_^−^: bicarbonate ion; Ti/Ttot: inspiratory duty cycle; *V*ex: tidal volume. The 10-point modified Borg Scale, with 0 corresponding to no dyspnea and 10 corresponding to maximal dyspnea was used to evaluate the exertional dyspnea. Among the four Global Initiative for Chronic Obstructive Lung Disease (GOLD) stages, despite the different breathing patterns ((**a**): *f*_R_, (**b**): *V*ex, and (**c**): Ti/Ttot) during exercise, patients with COPD did not regulate the exertional acidosis ((**d**): HCO_3_^−^) to stop exercise, reaching a similar exertional dyspnea level (**e**). Using the Kruskal-Wallis test to compare the groups consisting of the four GOLD stages, there was a significant difference in *f*_R_ (*p* = 0.0021), *V*_T_ (*p* < 0.0001), Ti/Ttot (*p* < 0.0001), and HCO_3_^−^ (*p* < 0.0001) at peak exercise, and HCO_3_^−^ (*p* < 0.0001) at rest. Using the Steel-Dwass test to carry out between-group comparisons, *a, *p* < 0.05 versus GOLD I; *b, *p* < 0.05 versus GOLD II; *c, *p* < 0.05 versus GOLD III. This is an original figure (no permission is required).

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
