# Peer review of "Motor Pathophysiology Related to Dyspnea in COPD Evaluated by Cardiopulmonary Exercise Testing"

_diagnostics, 2021, doi:10.3390/diagnostics11020364_

Round 1

Reviewer 1 Report

The current manuscript by Dr. Miki reviews the aspects of exertional dyspnea evaluation in COPD patients by cardiopulmonary exercise testing (CPET).

The article first briefly introduces the COPD disease and CPET, followed by discussions on exercise related factors focusing on COPD patients; this second section discusses cardiovascular disorders associated exertional dyspnea, the critical roles of lower limb fatigue and ventilator impairment and suggestions to reduce exertional dyspnea.

The point of view of the current article is interesting. The flow of the article is also suitable to describe the subject properly. Also, the manuscript is overall well written, but minor editing will help greatly to convey the message more clearly.

Author should consider addressing the following suggestions:

  1. The review should start with more discussions on the COPD. Author describes COPD as “COPD is a disease linked to exposure to cigarette smoking and environmental pollutants”, which is simplified and does not address the key factors of disease CPET can measure. Also, COPD pathophysiology includes more etiological factors which should be discussed first before going into the actual topic of the article.
  2. The introduction section should briefly discuss the CPET components and measurement protocol. Author discusses CPET in terms of its possibility to be used in cardiovascular diseases, but an actual account on the processes is left out.
  3. Exercise Tolerance and Exercise-limiting Factors: “Predicting the level of daily activity based on resting pulmonary function alone is difficult”, please elaborate on this crucial statement.
  4. Cardiovascular diseases as COPD associated disorders are crucial, but author may consider avoiding detailed discussions heart diseases which veers away the attention from the main topic of the article.
  5. The figure 1 showing scheme on “Schematic pathophysiologic pathway exertional dyspnea in chronic obstructive pulmonary disease” is informative, but the mention of “Body –Gas exchange and transport” at the top is confusing. Please explain why the gas exchange panel should not be at the bottom of the schematic representation. Seems the proper flow should first indicate the lung system, cardiovascular system and muscle system, and finally its impairment on Gas exchange, followed by Exercise intolerance. Please comment.
  6. Figure 2 indicating responses by COPD patients to incremental exercise is crucial but the lack of scale on Y-axis makes it difficult to interpret. Please consider inclusion of proper scale to describe the components.
  7. A suggestion will be to include Borg scale as a figure to describe dyspnea and its association with COPD. This can be good choice of figure for the current article.
  8. Exertional Dyspnea Due to Ventilatory Impairment: Please consider providing a chemical aspect of exercise-induced acidosis and homeostasis of the blood. The critical role of this in Ventilatory Impairment requires a basic understanding of the key processes. The discussion will benefit greatly by a basic account of the process and the reasons of impairment.
  9. Improving Ventilatory Impairments to Reduce Exertional Dyspnea: “COPD is recognized as a disease that primarily obstructs the peripheral respiratory tract”, please note that COPD also includes lung parenchyma. And account of the involvement of pulmonary parenchyma in exertional dyspnea should be included.
  10. A brief discussion on Eastern medical practice acupuncture has been included with the suggestion of its beneficial effect on exertional dyspnea. This section should also include other non-conventional therapies to address exertional dyspnea. The mention of only acupuncture undermines other alternative therapies proposed to benefit patients of COPD and other related diseases.

Best wishes.

Author Response

Responses to reviewer comments

 Title: Motor pathophysiology related to dyspnea in COPD evaluated by cardiopulmonary exercise testing

Journal: Diagnostics
Submission ID: diagnostics-1071871

Responses to Reviewer 1’s comments

I am grateful to Reviewer 1 for the constructive comments and useful suggestions that have helped us to improve my paper considerably. As indicated in the responses that follow, I have taken all the comments and suggestions into account in the revised version of our paper. Please see the revised manuscript, in which the changes have been highlighted.

Reviewer 1’s comments:

Comments and Suggestions for Authors

The current manuscript by Dr. Miki reviews the aspects of exertional dyspnea evaluation in COPD patients by cardiopulmonary exercise testing (CPET).

The article first briefly introduces the COPD disease and CPET, followed by discussions on exercise related factors focusing on COPD patients; this second section discusses cardiovascular disorders associated exertional dyspnea, the critical roles of lower limb fatigue and ventilator impairment and suggestions to reduce exertional dyspnea.

The point of view of the current article is interesting. The flow of the article is also suitable to describe the subject properly. Also, the manuscript is overall well written, but minor editing will help greatly to convey the message more clearly.

Author should consider addressing the following suggestions:

Comment #1:

The review should start with more discussions on the COPD. Author describes COPD as “COPD is a disease linked to exposure to cigarette smoking and environmental pollutants”, which is simplified and does not address the key factors of disease CPET can measure. Also, COPD pathophysiology includes more etiological factors which should be discussed first before going into the actual topic of the article.

Response to Comment #1:

Thank you for your interesting comments. As you suggested, “COPD is a disease linked to exposure to cigarette smoking and environmental pollutants” was deleted. Instead, this point has been clarified in the Introduction as follows: “The most frequent and important complaint of patients with COPD is exertional dyspnea, the pathophysiology of which includes more etiological factors” (p. 3, l. 4 –6).

Comment #2:

The introduction section should briefly discuss the CPET components and measurement protocol. Author discusses CPET in terms of its possibility to be used in cardiovascular diseases, but an actual account on the processes is left out.

Response to Comment #2:

Thank you for your interesting comments. As suggested, a brief discussion of CPET components and the measurement protocol have been included in the Introduction of the manuscript. “In CPET, briefly, two type protocols are used, that is, a maximum (symptom-limited) incremental exercise test and a constant work rate exercise test; and many exertional parameters are included, all of which are calculated using the ventilation amount, and the concentration of oxygen and carbon dioxide during inspiration or expiration” (p. 3, l. 15 –19).

Comment #3:

Exercise Tolerance and Exercise-limiting Factors: “Predicting the level of daily activity based on resting pulmonary function alone is difficult”, please elaborate on this crucial statement.

Response to Comment #3:

Based on your suggestion, a more detailed description was necessary. Predicting the level of daily activity based on resting pulmonary function alone is difficult, because other exercise limiting factors, such as cardiovascular disorders and lower limb fatigue, are not evaluated. I have therefore added a more detailed description (p. 4, l. 16 –18).

Comment #4:

Cardiovascular diseases as COPD associated disorders are crucial, but author may consider avoiding detailed discussions heart diseases which veers away the attention from the main topic of the article.

Response to Comment #4:

Thank you for your interesting comment. As suggested, detailed descriptions about cardiovascular diseases have been deleted (p. 5, l. 19 –21).

Comment #5:

The figure 1 showing scheme on “Schematic pathophysiologic pathway exertional dyspnea in chronic obstructive pulmonary disease” is informative, but the mention of Body –Gas exchange and transport” at the top is confusing. Please explain why the gas exchange panel should not be at the bottom of the schematic representation. Seems the proper flow should first indicate the lung system, cardiovascular system and muscle system, and finally its impairment on Gas exchange, followed by Exercise intolerance. Please comment.

Response to Comment #5:

Thank you for your interesting comments. The structure of Figure 1 has been changed based on your comments and those of the other reviewer.

Comment #6:

Figure 2 indicating responses by COPD patients to incremental exercise is crucial but the lack of scale on Y-axis makes it difficult to interpret. Please consider inclusion of proper scale to describe the components.

Response to Comment #6:

The structure of Figure 2 has been changed based on your comments and those of the other reviewer.

Comment #7:

A suggestion will be to include Borg scale as a figure to describe dyspnea and its association with COPD. This can be good choice of figure for the current article.

Response to Comment #7:

Thank you for your interesting comment. I agree that adding information on the exertional dyspnea will improve the manuscript and have added a figure to describe exertional dyspnea in the new Figure 2.

Comment #8:

Exertional Dyspnea Due to Ventilatory Impairment: Please consider providing a chemical aspect of exercise-induced acidosis and homeostasis of the blood. The critical role of this in Ventilatory Impairment requires a basic understanding of the key processes. The discussion will benefit greatly by a basic account of the process and the reasons of impairment.

Response to Comment #8:

Thank you for your interesting comments. I agree that discussing the chemical aspects of exercise-induced acidosis and homeostasis of the blood is necessary. Interestingly, in patients with COPD who possess various resting pulmonary functions, ventilation limitations, and exercise tolerances, exercise was terminated when exercise-induced acidosis (pH) and dyspnea during maximal exercise were comparable, as shown in the new Figure 2. CO2 transport is affected by the exertional pH change accompanying CO2 production due to ventilatory impairments or lactate production during exercise. As recommended, the necessary description has been added to the manuscript (p. 9, l. 17 –19).

Comment #9:

Improving Ventilatory Impairments to Reduce Exertional Dyspnea: “COPD is recognized as a disease that primarily obstructs the peripheral respiratory tract”, please note that COPD also includes lung parenchyma. And account of the involvement of pulmonary parenchyma in exertional dyspnea should be included.

Response to Comment #9:

As you suggested, a description of pulmonary parenchyma in exertional dyspnea has been added to the manuscript (p. 10, l. 22 –24).

Comment #10:

A brief discussion on Eastern medical practice acupuncture has been included with the suggestion of its beneficial effect on exertional dyspnea. This section should also include other non-conventional therapies to address exertional dyspnea. The mention of only acupuncture undermines other alternative therapies proposed to benefit patients of COPD and other related diseases.

Response to Comment #10:

Thank you for your interesting comments. Although the mechanism of meditative movements (e.g., tai chi, yoga, and qigong) on COPD is unclear, a systematic review and meta-analysis reported that the meditative movements, as non-conventional therapies, might improve exercise capacity, dyspnea, and health-related quality of life in COPD patients. The necessary description has been added to the manuscript (p. 12, l. 21 –25).

Reviewer 2 Report

CPET is a notoriously important test in the diagnosis of cardiopulmonary diseases, including COPD, in particular to investigate the presence of dyspnea on exertion.

This article discusses the motor pathophysiology of COPD, which is directly related to the treatment of exertional dyspnea, using cardiopulmonary exercise testing (CPET).

Comments:

- the title is not very attractive, too general. It is suggested to to recall main purpose of investigating the motor pathophysiology of COPD in patient with exertional dyspnea using CPET

- it is suggested to review the abstract as there is no correspondence between it and the relevant points reported in the various paragraphs

- page 3, Paragraph “2.1. Exertional Dyspnea Due to Cardiovascular Disorders”. I would avoid writing “...In our study of patients with COPD ...”, it would be preferable to write "... in previous studies ..."

- the graphs shown in figure 2 are very simple and rudimentary. I recommend remaking the graphs in a precise and aesthetically more presentable way, or otherwise leaving the explanation of the same in the text

- also figure 1, personally I think it should be redone and made aesthetically more attractive.

- the conclusions are too synthetic and instead the significant points highlighted in the paragraphs should be summarized.

Author Response

Responses to reviewer comments

Title: Motor pathophysiology related to dyspnea in COPD evaluated by cardiopulmonary exercise testing

Journal: Diagnostics
Submission ID: diagnostics-1071871

Responses to Reviewer 2’s comments

I am grateful to Reviewer 2 for the constructive comments and useful suggestions that have helped us to improve my paper considerably. As indicated in the responses that follow, I have taken all the comments and suggestions into account in the revised version of our paper. Please see the revised manuscript, in which the changes have been highlighted.

Reviewer 2’s comments:

Comments and Suggestions for Authors

CPET is a notoriously important test in the diagnosis of cardiopulmonary diseases, including COPD, in particular to investigate the presence of dyspnea on exertion.

This article discusses the motor pathophysiology of COPD, which is directly related to the treatment of exertional dyspnea, using cardiopulmonary exercise testing (CPET).

Comments:

Comment #1:

The title is not very attractive, too general. It is suggested to to recall main purpose of investigating the motor pathophysiology of COPD in patient with exertional dyspnea using CPET

Response to Comment #1:

Thank you for your interesting comment. Based on your suggestion, the title has been changed to, “Motor pathophysiology related to dyspnea in COPD evaluated by cardiopulmonary exercise testing”.

Comment #2:

It is suggested to review the abstract as there is no correspondence between it and the relevant points reported in the various paragraphs

Response to Comment #2:

              Thank you for your interesting comments. The abstract has been changed to reflect the relevant points reported in the manuscript.

Comment #3:

Page 3, Paragraph “2.1. Exertional Dyspnea Due to Cardiovascular Disorders”. I would avoid writing “...In our study of patients with COPD ...”, it would be preferable to write "... in previous studies ..."

Response to Comment #3:

As recommended, the description has been changed to read, “In previous studies of patients with COPD…” (p. 5, l. 16-19).

Comment #4:

The graphs shown in figure 2 are very simple and rudimentary. I recommend remaking the graphs in a precise and aesthetically more presentable way, or otherwise leaving the explanation of the same in the text

Response to Comment #4:

Thank you for your important comments. Figure 2 has been extensively revised using the data of patients with COPD (GOLD I, II, III, and IV), as well as adding information related to exertional dyspnea.

Comment #5:

Also figure 1, personally I think it should be redone and made aesthetically more attractive.

Response to Comment #5:

As suggested, Figure 1 has been extensively revised to make it more esthetically pleasing and to improve readability.

Comment #6:

The conclusions are too synthetic and instead the significant points highlighted in the paragraphs should be summarized.

Response to Comment #6:

Thank you for your interesting comments. As suggested, the conclusions have been revised to incorporate your recommendations, as follows: “Diverse exertional dyspnea is related to the crosstalk between the heart, lungs, and muscles; further exploration of the exertional dyspnea patterns, which are related to cardiovascular disorders, ventilatory impairment, and/or lower limb fatigue, will not only facilitate elucidation of the dynamic pathophysiology of COPD but they will also will contribute directly to the treatment of patients with this disease. Reducing the air remaining in the lungs after the expiration of each exhaled breath is expected to improve dynamic hyperinflation, respiratory acidosis, and eventually dyspnea. Furthermore, reducing ventilatory demand or increasing oxygen utilization may also facilitate improvements in exertional dyspnea in COPD. Considering that CPET can provide key information on specific dysfunctions in COPD patients that can be used to help them maintain daily living activities and allow patients to feel that they can “walk with a little more ease”, staff providing respiratory care should make the most of CPET as a more approachable test.” (p. 13, l. 1-14).

Round 2

Reviewer 2 Report

The Authors followed the comments and suggestions of Reviewers. In particular Figures have been extensively revised to make it more esthetically pleasing and to improve readability. The same also applies to the abstract. I have no other comments to add